# Do Individual Differences in Perception Affect Awareness of Climate Change?

**DOI:** 10.3390/brainsci14030266

**Published:** 2024-03-09

**Authors:** Enrico Cipriani, Sergio Frumento, Simone Grassini, Angelo Gemignani, Danilo Menicucci

**Affiliations:** 1Department of Surgical, Medical, Molecular and Critical Area Pathology, University of Pisa, 56126 Pisa, Italy; enrico.cipriani@phd.unipi.it (E.C.); sergio.frumento@med.unipi.it (S.F.);; 2Department of Psychosocial Science, University of Bergen, 5015 Bergen, Norway; simone.grassini@uib.no; 3Clinical Psychology Branch, Azienda Ospedaliero-Universitaria Pisana, 56126 Pisa, Italy

**Keywords:** climate change, interoception, hygroreception, thermoception, environmental neuroscience, climate neuroscience, climate change perception, climate change awareness, climate skepticism, climate denial

## Abstract

One significant obstacle to gaining a widespread awareness of the ongoing climate change is the nature of its manifestations in relation to our perception: climate change effects are gradual, distributed, and sometimes seemingly contradictory. These features result in a lag in collective climate action and sometimes foster climate skepticism and climate denial. While the literature on climate change perception and belief has thoroughly explored its sociocultural and sociopolitical aspects, research on the potential contribution of psychophysiological factors remains scarce. In this perspective paper, we outline evidence and arguments for the involvement of psychophysiological systems such as thermoception, hygroreception, and interoception in modulating climate change awareness. We discuss psychophysiological mechanisms of climate change awareness in animals and humans, as well as possible sources of individual variance in climate change perception. We conclude by suggesting novel research questions which would be worthwhile to pursue in future studies.

## 1. Introduction

The United Nations Framework Convention on Climate Change (UNFCCC), in its Article 1, defines climate change as “[…] a change of climate which is attributed directly or indirectly to human activity that alters the composition of the global atmosphere and which is in addition to natural climate variability observed over comparable time periods.” [1]. Climate change is mostly a gradual phenomenon, geographically distributed, and sometimes seemingly contradictory in its manifestations [2]. These perceptual features of climate change and its effects are an obstacle to fostering awareness, understanding, and action toward climate change in the general population. Conscious awareness of climate change effects is critical to promoting pro-environmental behavior. The perceptual nature of climate change induces individuals to approach it in abstract terms or through ideological lenses [3]; for this reason, the sociocultural factors implicated in climate change perception have been thoroughly explored, while the scientific literature on the strictly psychophysiological aspects of the climate change processes remains scant. Raising temperatures and their consequences pose a threat to the neurobiological systems of both humans and animals [4,5], and novel psychopathological disorders are arising as a consequence of climate change, such as climate anxiety [6]. These health effects of climate change could benefit from a clearer picture of the involvement of psychophysiological systems.

In this perspective paper, we have gathered evidence and arguments for the possibility of a psychophysiological source of variance in climate change perception, with the intent of contributing to the body of literature on the determinants of belief in climate change [7,8,9]. While it is important to remark that gaining awareness of a phenomenon like climate change requires multiple stages of elaboration and the involvement of a diverse array of cognitive processes, including semantic recognition of a particular perception as “climate change”, the scope of this perspective paper is strictly focused on the early physiological stages of this process of gaining awareness. We discuss evidence for climate perception in animals, referencing the field of sensory ecology, and humans, by exploring the different sensory modalities through which humans interface with the climate: thermoception, hygroreception (i.e., perception of wetness/dryness), and interoception. We then discuss possible individual differences in sensory and perceptual processing that might influence a sizable share of climate change awareness, and we propose a series of programmatic research questions to stimulate further developments. While it is not a systematic review, this paper represents the initial attempt to exclusively review psychophysiological data concerning human perceptions of climate change.

## 2. Perception of Climate Change in Non-Human Animals

The survival fitness of organisms is highly dependent on available resources and environmental conditions. Organisms evolved their mechanisms to anticipate and respond to environmental changes, such as those occurring due to the alternation of seasons. Animals employ adaptive behaviors according to sensory cues gathered and processed from their environment; these sensory cues are the dominion of sensory ecology, the study of how animals use perceptual information in their lives and how sensory systems influence their evolution [10]. Animals react to environmental changes threatening their survival by deploying adaptive behaviors such as migration, hibernation, and encystment. We can infer the detection of climate change in animals by examining the modifications of these behavioral responses in association with phenomena associated with climate change [11], such as changing median temperatures, changes in humidity, and frequency of extreme weather events (for a comprehensive review on the impacts of climate change, we direct the reader to reports by the International Panel on Climate Change [12]).

In the specific case of migration, for example, while this behavior is highly dependent on non-climate related phenomena, such as variations in photoperiod due to the alternations of seasons [13,14], it is still subjected to plasticity due to weather conditions [15,16]. Changes in temperatures and humidity seem to be the strongest climate-related triggers of migration in many taxonomic groups (e.g., [17,18]) Moreover, non-anthropogenic climate phenomena such as the North Atlantic Oscillation (NAO) and the El Niño-Southern Oscillation Index (ENSO), have been linked to migratory behaviors (for a review, see [19]). Indeed, migration onset, which is in part driven by spring temperatures in migratory birds, is altered as a consequence of climate change [20]; an increase in median temperature has resulted in behavioral change, with earlier departures [21].

In this context, the effects of climate change can be conceived as a form of sensory pollution, that is, how anthropogenic stimuli interact with the senses of other organisms [22,23]. While less studied than sensory pollution due to artificial lights, noise, or synthetic chemicals [23], the sensory pollution introduced by climate change affecting thermoception, hygroreception, or indirect effects of global warming might introduce strong, unexpected effects on animal behavior. Sensory pollution due to climate change may also manifest in temperature-driven second-order effects on other sensory modalities, such as olfaction [24].

Animal adaptation to their environment can take the form of highly specialized sensory and physiological systems that humans may not possess (e.g., arthropods such as insects and arachnids possess multiple types of “hygroreceptors”—specific receptor cells for the detection of variation in humidity in the environment—which are absent in other taxonomic groups [25]). In the case of thermoception, many snake species have the ability to detect temperature differences in their environment using infrared radiation. These animals possess a “pit organ”, consisting of a cavity with a thin membrane covered in nerve endings, which contain “Transient Receptor Potential Ankyrin 1” (TRPA1) temperature-sensitive ion channels [26]. Transient Receptor Potential (i.e., TRP) channels are a common feature of thermoreceptive cells in vertebrates, including humans, as we will discuss later [27].

Besides sensory pollution, climate change can influence adaptation behavior in animals through non-sensory physiological pathways [28]. Temperature changes impact the physiology of ectotherms, such as invertebrates and fishes. This susceptibility might alter their migratory behavior not through strictly sensory processes but through their physiology [20]. One such example is the effect of temperatures on migratory behaviors in North American salmon due to metabolic alteration [29,30]. In this case, reactions to climate change are driven by strictly physiological regulatory processes.

## 3. Sensory Perception of Climate Change in Humans

So far in our discussion, we have established that animals possess multiple systems to detect changes in climate and react to them and that these systems are rooted in their physiology and underpin complex behaviors; but what about humans? The current literature on climate change perception in humans emphasizes the importance of top-down processes (from higher cognition to lower levels of processing). Meaning the influence of beliefs, attitudes, norms, and ideologies in shaping climate change perception [8,9]. Within this context, a neuropsychological causal model for top-down modulation of perception has recently been proposed: the “Motivated Attention” model of climate change perception and action [31] describes how our personal stance on climate change diverts our attention to fixate on elements that align with our preconceptions. This dynamic ends up generating a positive feedback loop: noticing things that are in line with our ideas further increases the strength of our convictions; The motivated attention framework is convincing, and indeed it can explain many of the perceptual phenomena surrounding climate change, such as the fact that people with higher pre-existing knowledge of the phenomenon of climate change are more likely to notice its effects in everyday life.

Like other animal species, humans rely on sensory cues from their environment to adapt to environmental conditions and possess physiologically determined regulatory systems attuned to environmental cues and natural rhythms, such as circadian and ultradian cycles following variations in the daylight period [14,32]. Across the globe, human societies adopted sedentary or nomadic lifestyles. Both of these required attunements to environmental conditions to increase survival. On this note, there is an accumulating body of evidence suggesting the idea that the human brain is competent in detecting evolutionarily stimuli that resulted from evolutionary pressure, such as some specific phobias for natural threats, even on a pre-conscious level [33,34]. For example, visual stimuli possessing snake-like perceptual features modulate specific attention-related brain activities, even in the absence of subjectively perceived phobias toward the animals [35]. Moreover, sensory stimuli associated with safe, resource-rich natural environments, such as hearing birdsong, or smelling vegetation or rain, frequently evoke positive affective and physiological responses [36,37,38].

The most conspicuous effects of climate change are principally constituted by alterations in weather patterns in the form of altered precipitations or extreme weather events, such as heat waves or wet-bulb events [12]. The perception of these effects is underpinned by human sensory systems involved in sensing temperature (thermoception), and sensing humidity/dryness (hygroreception; see Figure 1).

## 4. Thermoception

Global warming, the principal driver of climate change, has led to an increase in average global temperatures, as well as an increase in the frequency of deadly heat waves, which constitute a critical health hazard [12]. Temperature regulates all biochemical reactions via thermodynamics; for this reason, humans and all other living beings possess thermoreceptive and thermoregulatory systems to maintain thermal homeostasis [39]. Human thermoception relies on thermoreceptor cells that convey alterations in environmental temperature via temperature-sensitive cation channels of the TRP (Transient Receptor Potential) family [27,40]. In particular, two channel proteins, “Transient Receptor Potential Melastin 8” (TRPM8) and “Transient Receptor Potential Vanilloid 1” (TRPV1), appear to be involved in cold and heat sensitivity, respectively. Menthol is an agonist for TRPM8, while Capsaicin is an agonist for TRPV1. These two compounds are appreciated as flavoring agents worldwide and are known for their capacity to evoke sensations of cold or heat.

TRPM8 and TRPV1 are mostly expressed in distinct “cold” and “hot” afferent neurons, whose cell body is localized in the Dorsal Root Ganglia (for thermoception on the body surface), and the Trigeminal Ganglia (for thermoception of face and head). Through the spinal cord, these neuronal populations project to the preoptic area of the hypothalamus, the somatosensory thalamic nuclei, and the parabrachial nuclei in the brainstem. These are thought to undercut the thermoregulatory, somatotopic, and affective/hedonic qualities of thermoception, respectively [41]. From these centers, thermal information is then relayed to the primary and secondary somatosensory cortices, and the anterior insular cortex [39].

Human thermoception appears to be able to detect both absolute temperature and temperature differences. However, while both “cold” and “warm” cells encode for absolute skin sensation, “cold” cells are often activated by dynamic cooling processes [38,40].

Since temperature alterations due to climate change are appreciable only on a scale of months or years, the perception of these temperature differences requires the reliance on memory for somatic states. However, this recollection is principally operated in reference to proximal information, such as the past few years, rather than historical temperature values [42].

In general, humans use readily available memory to assess temperature information on climate change, rather than more analytical forms of cognition [43]. Even current daily temperatures influence belief in temperature change due to global warming and willingness to act towards it [44], an effect known as the “local warming” effect. In addition, even the instantaneous visceral thermoreceptive state of an individual seems to influence the saliency of global warming as a problem, as evidenced by a study by Lewandowski and colleagues [45] in which they manipulated the visceral state of the participants by administering menthol flavored chewing gums, thus stimulating TRPM8 receptors.

The preponderance of readily available sensory information, as opposed to analytical, long-term memory recall and reasoning, increases the difficulty of associating one’s actions (e.g., CO_2_-releasing behaviors) and long-term global warming effects.

Indeed, the rules of contingency on which the most basic forms of learning are based clash with the extremely gradual nature of most climate change phenomena [46].

## 5. Hygroreception

Together with heat waves, climate change has led to an increase in “wet-bulb events” [12,47]. These are extreme weather phenomena in which the combination of temperature and relative humidity in the air impedes human thermoregulation, constituting a critical hazard for human health and survival, especially in urbanized environments in temperate and tropical regions [12].

There is no evidence for the existence of specialized human hygroreception systems [48]. Humans likely rely on multisensory integration to detect differences in wetness or humidity; tactile sensations are the primary drivers of wetness perception, with a considerable contribution from thermoception [48]. Wetness is conveyed to the spinal column via afferent myelinated Aβ fibers coding for mechanosensation and Aδ coding for cold thermosensation [48]. Molecular thermoreceptors for cold perception (e.g., TRPM8) have been implicated in the thermoreceptive aspect of wetness sensation [49].

Together with this initial multisensory integration, other sensory systems contribute to the feeling of wetness: specific patterns of sound can be interpreted by the brain as signifying dryness or wetness, as evidenced by the “parchment–skin illusion”. The illusion can be evoked by asking participants to hear a pitch-shifted recording of their hands rubbing one against the other. When the pitch was increased to higher frequencies, participants reported that the sound of their hands felt significantly drier, “like parchment paper” [50]. Vision also contributes to the perception of wetness: objects can be perceived as either wet or dry entirely based on their visual properties, such as variations in color and brightness [51]. Moreover, visual stimuli, in combination with electrotactile stimulation, are capable of producing convincing wetness effects in virtual environments [52].

Another intriguing possibility for wetness perception in humans might come through olfaction; a common experience after rainfall is the perception of “petrichor”, a musty, earthy smell, often associated with pleasant affective states and relaxation [53,54]. The smell of petrichor is induced by the inhalation of geosmin, a metabolite of some microorganisms, especially cyanobacteria. When these organisms die in dry periods, large quantities of geosmin are deposited in the soil [55]. When the soil is disturbed, such as during precipitations, geosmin is liberated into the air in aerosol form. Humans are extremely sensitive to concentrations of geosmin in the air, with estimations going as low as 9.5 parts per trillion [56]. Evidence also shows that smelling this compound is capable of inducing psychophysiological states of relaxation, even in the absence of other wetness sensory cues [36].

## 6. Interoception

Besides external temperature and wetness, mounting evidence suggests that interoception appears to play a significant role in shaping climate change perception. “Interoception” is a broad term encompassing sensory systems that relay information from the internal milieu of the body, unlike others that gather information from the external environment. Interoception includes visceral sensations like hunger and pain and plays a vital role in maintaining homeostasis [57]. Internal sensory information reaches the central nervous system (CNS) through two major afferent pathways: a vagal pathway, and a spinal pathway. These connections reach the brainstem nuclei, like the parabrachial nucleus (PBN) and the solitary tract nuclei (NTS). These in turn make contact with cortical areas, which include the somatic sensory cortex, the insula, the anterior cingulate cortex, and the prefrontal cortex [58].

Risen and Critcher demonstrated in a series of six experimental studies that manipulating visceral states increases temperature and dryness assessment, as well as belief in global warming [59]. The authors argue that current visceral states create a bias in assessment which favors judgments that are congruent with the current interoceptive state, in what they call the “visceral fit” effect. In their paper, individuals subjected to higher temperatures (both indoors and outdoors) were more likely to endorse climate change. Interestingly, this effect was independent of the political orientation of the individuals and influenced right-wing and left-wing individuals in the same manner. Similarly, participants to whom a state of thirst was previously induced rated the risk of drought and water scarcity as higher than those who were not thirsty [59]. In addition to the aforementioned study by Lewandowski and colleagues [45], it appears that current interoceptive states do indeed influence the perception of climate change.

It is important to note that, besides sociocultural variables like those mentioned in the previous sections, psychophysiological perceptual systems are influenced by numerous, non-sensory factors, such as mood, emotions [60], and physiological processes (e.g., thirst, see the “Interoception” section later in the text). Emotions and mood states can affect low-level perceptual processing in multiple ways [60], including affective states presenting as sensory phenomena or visceral sensations (e.g., shortness of breath due to anxiety). We could speculate that affective states can modulate climate perception through the aforementioned “visceral fit”. Indeed, mounting evidence shows that affective states, particularly negative ones, play an important role in shaping climate perception and action (for a review, see [61]).

The interoceptive sensation of pain also plays a role in climate perception; pain is integrated mainly with thermoception [62]. This is evidenced by the classic cold pressor test, used to evoke a pain response by hand immersion in containers filled with water at different temperatures [63], with many TRP receptors also serving as pain receptors [64]. An association between chronic pain exacerbation and weather changes has been established for millennia, and it has been experimentally verified [65]. However, a clear mechanistic explanation for this phenomenon remains to be established [66]. In addition, there seems to be a neurological cross-talk between the sensation of pain and thirst which might constitute a further avenue through which a climate-related stimulus can exert modulation through interoception [67,68].

Considering these lines of evidence, as well as the other instances of interaction and superposition of sensory systems that we mentioned so far in our discussion, it appears plausible that the perception of climate change effects arises as a result of multisensory integration, rather than single-sensory channels. While these sensory systems are shared by every human, a significant variance in the subjective perception of climate change still exists. Besides variance due to sociocultural factors, in the next section, we illustrate some likely sources of variation in the climate change perception of a psychophysiological nature.

## 7. Psychophysiological Individual Differences and Climate Change Perception

There are many possible sources of psychophysiological variations in climate change perception. These can intervene at every step of the perceptual chain, from genetic polymorphisms to individual differences in multisensory integration. Humans exhibit a wide range of individual sensitivity for both temperature and pain [69,70]. Genetic variants of genes coding for TRP channels correlate with individual differences in sensitivity to heat [71], and several TRP-coding gene polymorphisms have been associated with both heat sensitivity and chronic pain conditions (Table 1). Variants of the capsaicin-sensitive TRPV1, like the single nucleotide polymorphism (SNP) rs8065080, are associated with variations in sensitivity to hot and cold pain [72]. Individuals homozygous for this allele also show remarkable sensitivity to heat which is not reduced by the application of capsaicin [73]. Meanwhile, rs57716901 and rs61387317 SNP variants are associated with burning pain sensitivity [74]. Individuals carrying another polymorphism for the TRPV1 gene display altered tolerance to thermal pain, which is also moderated by personality traits [75]. Genetic variants of the TRPM8 cold receptor gene are particularly notable: these polymorphisms not only are associated with sensitivity to cold and painful cold [76,77], but their prevalence in populations is associated with local climate and average winter temperatures [78,79]. In particular, the prevalence of the rs17862920 variant appears to follow a latitude gradient, with lower concentrations around the equator (5% in Nigerian samples), and higher towards the pole (88% in Finnish samples) [78]. We might speculate that these variants constituted an advantageous sensory adaptation to specific climates that increased population fitness.

Besides structural genetic variants, epigenetic alterations in promoters of TRP channels, such as methylation, have also been associated with pain sensitivity [81]. Findings such as these might translate into the possibility that a degree of genetic plasticity exists in climate change perception; increasing adaptive pressure towards climate change effects might push for adaptations that increase individual sensitivity to alterations in climate. Indeed, epigenetic adaptive and maladaptive alterations are known to arise as a consequence of heat exposure and heat stress [82]. It might be possible that populations that have been subjected to extreme weather events related to climate change, such as heat waves or flooding, may carry epigenetic markers on genes related to thermoregulation.

There is still no evidence of an association between individual differences in interoceptive dimensions and perception of climate change. However, a study shows that individuals who practice techniques that improve interoceptive awareness, such as mindfulness meditation [83], possess higher levels of belief in climate change factors [84]. Although this effect might be ascribable to personality variables, we might speculate that the practice of mindfulness in itself may induce a sensitization effect. These contemplative techniques, when practiced consistently, are known to induce structural psychophysiological changes, such as alterations in brain connectivity with the insula, a major interoceptive hub [85].

Some individual differences in attentional styles and variables have already been associated with climate change attitudes and perceptions. For example, individuals possessing a global-to-local attentional style, as measured by the Navon task, tend to hold higher beliefs in the existence of climate change [86]. This individual preference for global or local perceptual styles has been associated with differences in multisensory integration, which might bias attention either towards detail or towards a more holistic perception [87].

## 8. Conclusions and Further Research

On the basis of the scientific findings presented in our paper, we propose several research questions that deserve further exploration in future studies (Table 2):


*#1: Do populations carrying TRP gene variants experience climate change differently?*


Research on these gene variants could reveal vulnerability factors for disorders that arise from climate change distress. Addressing these vulnerabilities could aid in preventing and treating these disorders.


*#2 Are there evolutionarily determined climate change sensory cues in humans?*


Given the specificity of the sensory effect of some stimuli (such as geosmin molecules), do they constitute innate, evolutionarily driven adaptations towards climate change? Do individuals experiencing climate anxiety or other climate-related affective states have a higher degree of sensitivity to these environmental cues? Besides increasing awareness, these stimuli might be employed as part of novel supporting therapies for distress and psychopathological disorders due to climate change.


*#3 What is the degree and speed of climate change that is necessary to happen so that humans can become aware of it?*


Arching back to the boiled frog apologue, it would be worthwhile determining something similar to the Just Noticeable Difference (JND; [88]) for a given climate change stimulus that would make an individual evaluate it as anomalous. For example, is there a minimal temperature difference that can be detected as out-of-season, or as caused by climate change? Individuating such a threshold could help in crafting awareness campaigns pointing out specific anomalies as they happen.


*#4 Can individual differences in perceptual awareness of climate change be measured?*


While differences in climate change beliefs and attitudes have been studied consistently, and are backed by validated psychometric scales, perceptual awareness of climate change remains to be measured in the general population. Developing a specific psychometric scale for this purpose might be worthwhile to investigate this aspect of climate change perception.


*#5 How does interoception affect climate perception?*


Much like research on genetic factors, research on the involvement of interoceptive factors in climate change perception could open new avenues for body awareness-based treatment applicable to emerging psychopathological disorders like climate anxiety. In this sense, the development of novel specialized mindfulness-based protocols could be promising. In addition, exploring interoceptive correlates of decision-making—such as the somatic marker hypothesis—in the context of sustainability-related choices, may uncover new avenues of research to foster pro-environmental decisions in everyday life.


*#6 How are the objective sensory data and subjective perceptions related to climate change represented in the brain?*


So far, we have detailed how receptors and low-level perceptual processing can interface with climate change. However, it remains to be investigated how the multisensory integration underlying the perception of climate change manifests itself on a cortical level. Are there specific neural signatures—such as local activations or functional networks—for the perception of climate change effects? On a similar note, it remains to be determined how top-down processes—such as motivated attention—modulate the perception of climate change on a neural level. This can help elucidate how variables ascribable to subjectivity influence the objective factors in climate change perception.

To conclude, we believe that this perspective paper highlights the need to open new avenues of research based on the scientific literature from different disciplines that have been linked here. We underline that further studies are necessary to ascertain whether individual differences in low-level receptors, psychophysiological systems, and cognitive styles can result in differences in climate change perception and whether this has any bearing on attitudes and behaviors relating to climate change. Furthermore, we believe that connecting the physiological knowledge of circadian and ultradian rhythms to that of phenology (i.e., the study of cyclical environmental phenomena [89]) and exploring human–environment interactions in the framework of sensory ecology and sensory pollution might be pivotal to expanding the knowledge of human awareness and reaction to climate change. Pursuing these new avenues might unveil new tools for engagement in climate action, and to improve adaptation, well-being, and healthcare in the changing climate.

## Figures and Tables

**Figure 1 brainsci-14-00266-f001:**
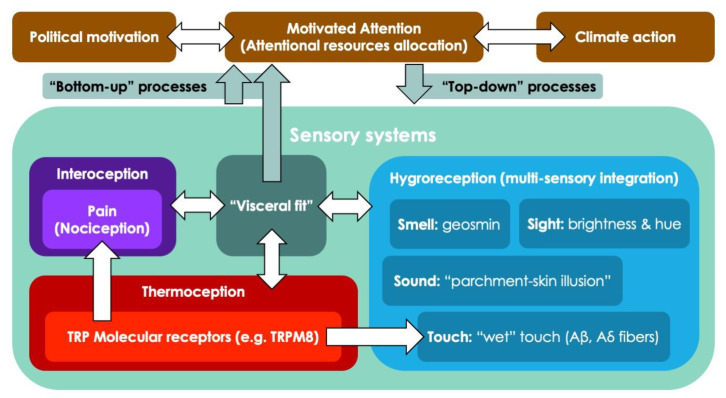
Model for the psychophysiological factors affecting climate change perception in humans. Sensory systems such as interoception, thermoception, and hygroreception interact with each other and share physiological elements (such as TRP receptors). Sensory systems influence climate change awareness through bottom-up processes, such as visceral fit. Sensory systems are themselves influenced by top-down processes, such as the allocation of attentional resources due to motivated attention.

**Table 1 brainsci-14-00266-t001:** TRP Genetic variants associated with sensitivity.

Type	SNP Code	Findings	Ref.
TRPM8	rs10166942	Correlated with geographical latitude.	[78]
TRPM8	rs10166942[C]	More prevalent in hotter climates. Decreases TRP8 expression. Reduced sensitivity to cold.	[76]
TRPM8	rs10166942[T]	More prevalent in colder climates.	[76]
TRPM8	rs11562975	Heterozygous individuals show an increased sensitivity to cold.	[77]
TRPM8	rs12992084	Association with cold pain sensitivity.	[79]
TRPM8	rs17862920	Allelic correlation with average winter temperatures.	[80]
TRPM8	rs7577262	Allelic correlation with average winter temperatures.	[80]
TRPV1	rs57716901	Associated with burning pain sensitivity	[74]
TRPV1	rs61387317	Associated with burning pain sensitivity	[74]
TRPV1	rs8065080	Cold hypoalgesia. Less heat hyperalgesia. Less pinprick hyperalgesia. Mechanical hypoesthesia.	[72]
TRPV1	rs8065080	After capsaicin application: Less warm-detection in heterozygotes/WT. Gain in heat-pain sensitivity in heterozygotes/WT.	[73]

SNP: Single nucleotide polymorphism; WT: wild type.

**Table 2 brainsci-14-00266-t002:** Further research questions.

N.	Research Question
#1	*Do populations carrying TRP gene variants experience climate change differently?*
#2	*Are there evolutionarily determined climate change sensory cues in humans?*
#3	*What is the degree and speed of climate change that is necessary to happen so that humans can become aware of it?*
#4	*Can individual differences in perceptual awareness of climate change be measured?*
#5	*How does interoception affect climate perception?*
#6	*How are the objective sensory data and subjective perceptions related to climate change represented in the brain?*

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
