# Peer review of "Do Individual Differences in Perception Affect Awareness of Climate Change?"

_brainsci, 2024, doi:10.3390/brainsci14030266_

Round 1

Reviewer 1 Report

Comments and Suggestions for Authors

This is a worthwhile manuscript that focuses on possible individual differences in climate perception between animals and humans and whether this affects climate change awareness. Overall the paper has a reasonable theme, but still has some issues and shortcomings that should be discussed, as follows:

1. there is a lack of further data to support an increase in the average rate of movement of large herbivores prior to the onset of rainfall patterns in the perception of climate change by animals, and a lack of generalization.

2. the lack of theoretical evidence that changes in temperature and humidity are the most strongly correlated factors with climate.

3. the conclusion that senses are localized from behavioral changes in species living in northern latitudes and equatorial regions lacks some generalization

4. the human perception of climate change does not take into account factors that affect the psychophysiological perception of human beings

5. the lack of experimental validation of possible sources of psychophysiological differences in climate change perception

6. the authors should highlight the methodology used to demonstrate the interaction between receptors and low-level perceptual processing and climate change.

7. the authors should emphasize the novelty of the paper and add theoretical justifications where appropriate.

Comments on the Quality of English Language

The overall English quality is good, just some individual sentences' grammar need moderate correction.

Author Response

We thank you for the stimulating comments and concerns, which we have taken into the utmost consideration and which we believe have led to improvements. 

In the following, we report point by point the issues you raised along with our response. We answered each question, and we modified the manuscript accordingly.

This is a worthwhile manuscript that focuses on possible individual differences in climate perception between animals and humans and whether this affects climate change awareness. Overall the paper has a reasonable theme, but still has some issues and shortcomings that should be discussed, as follows:

Authors: We thank the Reviewer for the kind words, and for thoroughly examining the manuscript. We respond to the point raised as follows:

  1. there is a lack of further data to support an increase in the average rate of movement of large herbivores prior to the onset of rainfall patterns in the perception of climate change by animals, and a lack of generalization.

Authors: We agree that the paper cited constitutes isolated evidence. We thus opted to remove its mention.

  1. the lack of theoretical evidence that changes in temperature and humidity are the most strongly correlated factors with climate.

Authors: We agree that we only implied this logical step in our argumentation. To bridge this logical step, we added a sentence on the link between climate change, temperature, and humidity, citing the latest IPCC report on climate change. We also increased the number of references and changed the phrasing to not use the word “strongest”, which might have been misleading for the reader: 

 “We can infer the detection of climate change in animals by examining modifications of these behavioral responses in association with phenomena associated with climate change [10], such as changing median temperatures, changes in humidity, and frequency of extreme weather events (for a comprehensive review on the impacts of climate change, we direct the reader to reports by the International Panel on Climate Change [11]).”

  1. the conclusion that senses are localized from behavioral changes in species living in northern latitudes and equatorial regions lacks some generalization.

Authors: We agree with the Reviewer that the conclusion lacks evidence. We decided to remove this paragraph since doing so does not detract elements from our paper.

  1. the human perception of climate change does not take into account factors that affect the psychophysiological perception of human beings.

Authors: We thank the Reviewer for raising this issue. We added a paragraph in the revised text to address this, particularly the contribution of emotions and affective processes in shaping perception: 

“It is important to note that, besides sociocultural variables like those mentioned in the previous sections, psychophysiological perceptual systems are influenced by numerous, non-sensory factors, such as mood, emotions [59], and physiological processes (e.g. thirst, see the “Interoception” section later in the text). Emotions and mood states can affect low-level perceptual processing in multiple ways [59], including affective states presenting as sensory phenomena or visceral sensations (e.g., shortness of breath due to anxiety). We could speculate that affective states can modulate climate perception through the aforementioned “visceral fit”. Indeed, mounting evidence shows that affective states, particularly negative ones, play an important role in shaping climate perception and action (for a review, see [60]).”

  1. the lack of experimental validation of possible sources of psychophysiological differences in climate change perception.

Authors: We agree that this section of our manuscript was lacking. We expanded the section, including more experimental evidence on individual differences.

  1. the authors should highlight the methodology used to demonstrate the interaction between receptors and low-level perceptual processing and climate change.

Authors: We agree with the Reviewer that research on the links between these elements is necessary. However, the experimental literature on this topic remains scarce. We address this research gap programmatically in our “Conclusions and further directions” section of this perspective paper, inviting further research on the subject. In parallel, to bolster our argumentation, we expanded the section on individual differences, providing more evidence on genetic variants, perceptual individual differences, and adaptation to climate.

  1. the authors should emphasize the novelty of the paper and add theoretical justifications where appropriate.

Authors: We thank the Reviewer for pointing out the novelty of this work. To emphasize it, we amended the introduction section of our paper: 

“To our knowledge, this paper is the first to review strictly psychophysiological data on climate change perception in humans.”

Reviewer 2 Report

Comments and Suggestions for Authors

The author starts with the interesting topic of boiling frogs in warm water and reviews existing research on the relationship between climate change and psychology and physiology. The paper has certain significance, but there are many areas that need to be improved:

1.The research conclusion is not clear. Does the question mentioned in the title of the paper exist? The author did not provide a clear research conclusion.

2. The author also discussed animals in the article. Is it necessary to do so? Because psychological problems only exist in humans.

3. The lack of charts in the paper results in less intuitive research processes and conclusions.

4. The parentheses at the end of the second to last paragraph on page 3 mention "See Figure 1", but in the paper, I did not see Figure 1, and there is a caption similar to Figure 1 below. I suspect the author may have forgotten to include Figure 1 in the paper.

Given the above reasons, I suggest that the author make major revisions before resubmitting.

Author Response

We thank you for the stimulating comments and concerns, which we have taken into the utmost consideration and which we believe have led to improvements. 

In the following, we report point by point the issues you raised along with our response. We answered each question, and we modified the manuscript accordingly.

The author starts with the interesting topic of boiling frogs in warm water and reviews existing research on the relationship between climate change and psychology and physiology. The paper has certain significance, but there are many areas that need to be improved:

Authors: We thank the Reviewer for the kind words expressed about the themes and the significance of our paper, and for thoroughly examining the manuscript. We respond to the points raised as follows:

  1. The research conclusion is not clear. Does the question mentioned in the title of the paper exist? The author did not provide a clear research conclusion.

Authors: We agree with the Reviewer that the conclusions of our paper deserve more clarity. We included a specific “Conclusions and further directions” section at the end of our paper.

  1. The author also discussed animals in the article. Is it necessary to do so? Because psychological problems only exist in humans.

Authors: We thank the Reviewer for raising this issue. A section concerning animals may appear unnecessary, however, we believe that discussing how animals detect and react to environmental changes, and physiologically-determined reactions can help further the understanding of the physiological systems of humans, and how they might operate in comparison. Moreover, this section serves to introduce two important concepts: sensory ecology and sensory pollution. We believe that connecting the physiological knowledge of circadian and ultradian rhythms to that of phenology (i.e. the study of cyclical environmental phenomena, like migrations) and exploring human-environment interactions in the framework of sensory ecology and sensory pollution, might be pivotal to expanding the knowledge of human awareness and reaction to climate change. 

  1. The lack of charts in the paper results in less intuitive research processes and conclusions.

Authors: We agree that charts, tables, and diagrams might enhance the readability of our paper. We added a table summarizing genetic variants associated with temperature sensitivity, and a table summarizing the research questions we outlined in the “conclusions and further directions” section, to improve readability.

  1. The parentheses at the end of the second to last paragraph on page 3 mention "See Figure 1", but in the paper, I did not see Figure 1, and there is a caption similar to Figure 1 below. I suspect the author may have forgotten to include Figure 1 in the paper.

Authors: We apologize for the inconvenience. Other Reviewers have reported the image missing from the review version of the paper. Since the figure is present both in the original manuscript we submitted to the site, and the downloadable draft version available in MDPI’s portal, we believe that there must have been a technical problem resulting in the image not appearing in the version of the manuscript that was sent to the Reviewers. We attach to this message a .png version of the figure for your viewing. We apologize again for the inconvenience.

Reviewer 3 Report

Comments and Suggestions for Authors

This study is very interesting. I enjoyed reading this from an empathetic perspective. However, this study lacked of scientific empirical research and analysis. Although the authors cited a lot of previous papers to try to explain and support their own opinions, there is still too much conjecture in such an inference and lack of support from research evidences. I think the questions at the end of this article is very inspiring for brain science, but I suggest the authors transfer this article to submit to journals in the "Report" category. 

Author Response

We thank you for the stimulating comments and concerns, which we have taken into the utmost consideration and which we believe have led to improvements. 

In the following, we report point by point the issues you raised along with our response. We answered each question, and we modified the manuscript accordingly.

This study is very interesting. I enjoyed reading this from an empathetic perspective. However, this study lacked of scientific empirical research and analysis. Although the authors cited a lot of previous papers to try to explain and support their own opinions, there is still too much conjecture in such an inference and lack of support from research evidences. I think the questions at the end of this article is very inspiring for brain science, but I suggest the authors transfer this article to submit to journals in the "Report" category.

Authors: We thank the Reviewer for the positive words and the interest expressed in our paper. Concerning the lack of empirical data, we have to point out that this contribution has been submitted as a perspective article, and therefore, a measure of inference and personal conjecture is both allowed and necessary to provoke further developments in empirical research. However, the issue raised by the Reviewer is indeed a limitation, and the need for empirical research has been stressed in the final sections of our revised paper.

Reviewer 4 Report

Comments and Suggestions for Authors

Review

This article examines the evidence and arguments for involving psychophysiological systems such as thermoception, hygroception, and interoception in modulating climate change. It is a very interesting text. However, I suggest some changes to make the text more impactful and contribute to this interesting area of study.

Title.

As far as the title is concerned, it would be a lot more interesting to raise it directly. I would remove "Frogs jumping off the pot".

This is a new and interesting topic. It might have more impact if the title was more specific.

Introducction

The first part of the introduction is very broad and provides a context that, although correct, could make the reader lose the meaning of the text. Therefore, I suggest that the first part (paragraphs) be shortened and made more concrete, directly addressing the problem from the psychophysiological and health point of view, as proposed by the authors.  It is also necessary that the introduction, before dealing with each of the topics, anticipates the results of the effect of psychophysiological systems, e.g. citing a number of studies that anticipate what the reader will find in each of the sections of the introduction.  It is important that authors clarify the aim/hypothesis of the article and the type (review, perspective, etc.) of the paper in the last paragraph.

Perception of climate change in non-human animals

Although the authors provide evidence that non-human animals perceive climate change, I propose to add and deepen the psychophysiological or biological mechanisms that allow non-human animals to perceive climate change.

The text of the paper does not include Figure 1.

Interoception.

As with the other processes, it is important to include a section that explains the psychophysiological or biological mechanisms of interoception.

Psychophysiological individual differences and climate change perception: possible research targets

In this section, it would be important to include more explicit evidence (if any) on individual differences in thermo- and hygroperception.

A session on conclusions and future directions should be added to the text. This will make the idea of the paper more concrete. It will also make the message to be sent much more effective.

Comments on the Quality of English Language

Could make minor improvements

Author Response

We thank you for the stimulating comments and concerns, which we have taken into the utmost consideration and which we believe have led to improvements. 

In the following, we report point by point the issues you raised along with our response. We answered each question, and we modified the manuscript accordingly.

This article examines the evidence and arguments for involving psychophysiological systems such as thermoception, hygroception, and interoception in modulating climate change. It is a very interesting text. However, I suggest some changes to make the text more impactful and contribute to this interesting area of study.

Title.

As far as the title is concerned, it would be a lot more interesting to raise it directly. I would remove "Frogs jumping off the pot".

This is a new and interesting topic. It might have more impact if the title was more specific.

Authors: We agree that the colorful mention of frogs in the title, although enticing, might be misleading and a source of confusion. We propose a more direct and concise title.

Introduction

The first part of the introduction is very broad and provides a context that, although correct, could make the reader lose the meaning of the text. Therefore, I suggest that the first part (paragraphs) be shortened and made more concrete, directly addressing the problem from the psychophysiological and health point of view, as proposed by the authors.

Authors: We agree that the introduction deserves to be more to the point. In line with the alterations made to the title, we removed the mention of the boiling frogs’ story, thus starting directly with the initial argumentation. 

It is also necessary that the introduction, before dealing with each of the topics, anticipates the results of the effect of psychophysiological systems, e.g. citing a number of studies that anticipate what the reader will find in each of the sections of the introduction.

Authors: We concur with the Reviewer that an anticipation of the contents of the article is a necessary part of an introductory section, we expanded the introduction to do so:

“We will discuss evidence for climate perception in animals, referencing the field of sensory ecology, and humans, by exploring the different sensory modalities through which humans interface with the climate: thermoception, hygroreception (i.e. perception of wetness/dryness), and interoception. We will then discuss possible individual differences in sensory and perceptual processing that might influence a sizable share of climate change awareness, and we will propose a series of programmatic research questions to stimulate further developments.”

It is important that authors clarify the aim/hypothesis of the article and the type (review, perspective, etc.) of the paper in the last paragraph.

Authors: We thank the Reviewer for the suggestion to write the aims/hypothesis of the article and its type in the last paragraph of the introduction. The revised paper contains an explicit indication of the article type and the aims of the paper in the last paragraph of the Introduction section.

Perception of climate change in non-human animals

Although the authors provide evidence that non-human animals perceive climate change, I propose to add and deepen the psychophysiological or biological mechanisms that allow non-human animals to perceive climate change.

Authors: we agree with the Reviewer that this topic deserves a more in-depth discussion, however, other Reviewers asked us to remove material from this section and not to enlarge it. We propose thus not to broaden the scope of this section for the sake of balance.

The text of the paper does not include Figure 1.

Authors: We apologize for the inconvenience. Other Reviewers have reported the image missing from the review version of the paper. Since the figure is present both in the original manuscript we submitted to the site, and the downloadable draft version available in MDPI’s portal, we believe that there must have been a technical error resulting in the image not appearing in the version of the manuscript that was sent to the Reviewers. We attach to this message a “.png” version of the figure for your viewing. We apologize again for the inconvenience.

Interoception.

As with the other processes, it is important to include a section that explains the psychophysiological or biological mechanisms of interoception.

Authors: We agree with the Reviewer that the theoretical basis of interoception requires an explanation. We expanded this section to add further context.

Psychophysiological individual differences and climate change perception: possible research targets

In this section, it would be important to include more explicit evidence (if any) on individual differences in thermo- and hygroperception.

Authors: We thank the Reviewer for requesting further evidence. We revised the text to add further research on individual differences in thermoception.

A session on conclusions and future directions should be added to the text. This will make the idea of the paper more concrete. It will also make the message to be sent much more effective.

Authors: As kindly suggested by the Reviewer, we added a “Conclusions and further directions” section to our paper to increase the concreteness and efficacy of the paper.

Reviewer 5 Report

Comments and Suggestions for Authors

 The paper is good. Some major and minor comments are presented below:

1. Minor: Please correct use of your abbreviations, e.g. TRP. Please introduce abbreviations clearly.

2. Minor: Please reconsider one-sentence paragraphs as they are unwanted.

3. Minor: Please correct typos in the paper.

4. Major: Please create a table with summary of your review where you can describe (but shortly, in a synthesized manner) your key mechanisms/hypotheses/implications/research problems etc. This will increase significantly the readability and impact of your paper.

5. Major: Figure 1 is not presented in the manuscript. Please add your figure as all materials should be peer-reviewed. Please prepare attentively your materials before submitting as a lack of necessary materials increase review time and decrease the quality assessment of the paper.

Author Response

We thank you for the stimulating comments and concerns, which we have taken into the utmost consideration and which we believe have led to improvements. 

In the following, we report point by point the issues you raised along with our response. We answered each question, and we modified the manuscript accordingly.

The paper is good. Some major and minor comments are presented below:

Authors: We thank the Reviewer for the very positive reception of our manuscript. We respond to each point and comment raised as follows:

  1. Minor: Please correct use of your abbreviations, e.g. TRP. Please introduce abbreviations clearly.

Authors: We corrected the abbreviations used in the text, as pointed out by the Reviewer.

  1. Minor: Please reconsider one-sentence paragraphs as they are unwanted.

Authors: We amended the text by incorporating and rephrasing one-sentence paragraphs, as pointed out by the Reviewer.

  1. Minor: Please correct typos in the paper.

Authors: As requested, we performed a grammar check of the paper and removed all the typographic errors we could find.

  1. Major: Please create a table with summary of your review where you can describe (but shortly, in a synthesized manner) your key mechanisms/hypotheses/implications/research problems etc. This will increase significantly the readability and impact of your paper.

Authors: We thank the Reviewer for the suggestion: to increase readability, we added a table to summarize the genetic variants associated with heat sensitivity and the proposed research questions.

  1. Major: Figure 1 is not presented in the manuscript. Please add your figure as all materials should be peer-reviewed. Please prepare attentively your materials before submitting as a lack of necessary materials increase review time and decrease the quality assessment of the paper.

Authors: We apologize for the inconvenience. Other Reviewers have reported the image missing from the review version of the paper. Since the figure is present both in the original manuscript we submitted to the site, and the downloadable draft version available in MDPI’s portal, we believe that there must have been a technical error resulting in the image not appearing in the version of the manuscript that was sent to the Reviewers. We attach to this message a “.png” version of the figure for your viewing. We apologize again for the inconvenience.

Round 2

Reviewer 2 Report

Comments and Suggestions for Authors

Inquiries were carefully answered, and the novelty and effectiveness were clarified. Therefore, it was judged to be worth publishing.

Author Response

We thank the Reviewer for the kind words and wish them a happy new year.

Reviewer 3 Report

Comments and Suggestions for Authors

The authors had modified some contents of this manuscript, and mede their claims stronger and more clear. However, this manuscript is not a systematic review paper,  therefore, it is not possible to determine whether there is bias when judging this manuscript for readers. Besides, although the hypothesis of this article can provide some new ideas and imagination, it does not provide new evidence.I suggest the authors transfer this article to submit to journals in the "Report" category. 

Author Response

We thank the Reviewer for replying. We respectfully disagree with the Reviewer on this issue. We reassert the notion that this is a Perspective article, thus a systematic review of the literature goes beyond the scope of this publication category. Regarding potential bias, we believe that Perspective papers, as stated in MDPI's page on article types "Emphasis is placed on future directions of the field and on the personal assessment of the author". Authors' assessment – and the biases that come with it – are crucial aspects of this publication. 

Reviewer 5 Report

Comments and Suggestions for Authors

Dear Authors,

Thank you for your significant revision. The manuscript is structured and has a nice quality of presentation.

Only minor comments:

Please do not use "on the other" hand if you have no "on the one hand" before.

Please avoind using two closest brackets, e.g., see on page 4.

You could also add more keywords for better indexing your paper in scientific bases.

There are some typos (e.g., extra spaces), and I believe the authors will fix these. 

Author Response

We thank the Reviewer again for the positive reception of our paper. We respond to the minor comments raised:

Please do not use "on the other" hand if you have no "on the one hand" before.

Authors: We thank the Reviewer for pointing this out. We amended the text.

Please avoind using two closest brackets, e.g., see on page 4.

Authors: We thank the Reviewer for pointing this out. We amended the text.

You could also add more keywords for better indexing your paper in scientific bases.

Authors: We added four more keywords as suggested.

There are some typos (e.g., extra spaces), and I believe the authors will fix these. 

Authors: We thank the Reviewer for pointing this out. We amended the text and performed an extensive grammar check.

We thank the Reviewer again and wish them a happy new year.